# Robustness Analysis of Non-Convex Stochastic Gradient Descent using Biased Expectations

**Kevin Scaman**     **Cédric Malherbe**
Huawei Noah's Ark Lab

## Abstract

This work proposes a novel analysis of stochastic gradient descent (SGD) for non-convex and smooth optimization. Our analysis sheds light on the impact of the probability distribution of the gradient noise on the convergence rate of the norm of the gradient. In the case of sub-Gaussian and centered noise, we prove that, with probability $1 - \delta$, the number of iterations to reach a precision $\varepsilon$ for the squared gradient norm is $O(\varepsilon^{-2} \ln(1/\delta))$. In the case of centered and integrable heavy-tailed noise, we show that, while the expectation of the iterates may be infinite, the squared gradient norm still converges with probability $1 - \delta$ in $O(\varepsilon^{-p} \delta^{-q})$ iterations, where $p, q > 2$. This result shows that heavy-tailed noise on the gradient slows down the convergence of SGD without preventing it, proving that SGD is robust to gradient noise with unbounded variance, a setting of interest for Deep Learning. In addition, it indicates that choosing a step size proportional to $T^{-1/b}$ where $b$ is the tail-parameter of the noise and $T$ is the number of iterations leads to the best convergence rates. Both results are simple corollaries of a unified analysis using the novel concept of *biased expectations*, a simple and intuitive mathematical tool to obtain concentration inequalities. Using this concept, we propose a new quantity to measure the amount of noise added to the gradient, and discuss its value in multiple scenarios.

## 1   Introduction

Stochastic Gradient Descent (SGD) and its variants (Adam [1], RMSProp [2], or Nesterov's accelerated gradient descent [3]) are used in a wide variety of tasks to train Machine Learning models. Indeed, the scalability of many learning algorithms, such as support vector machines [4], logistic regression [5], Lasso [6] and more recently deep neural networks [7] essentially rely on the efficiency (and robustness) of stochastic optimization methods. However, one specificity of SGD is its inherent noise which originates either from the sampling of training points, the presence of noise in the gradients computation, or the shape of the target function [8, 9, 10]. While SGD is known to be robust in practice and its convergence behavior is well-understood in the convex setting [11, 12, 3, 13], many of its properties are not yet fully understood, and particularly in settings related to Deep Learning practice where gradients can be extremely noisy and the target function presents many local optima. In particular, settings with unbounded variance noise were recently shown to appear [14, 15] when training NLP models such as BERT [16] over large corpora, and vision models such as AlexNet [17] on Cifar10. As a result, SGD may present instabilities that are often solved by running the optimization multiple times, a technique refered to as multi-start. Recently, several authors explored these frameworks by adapting the tools developed in convex analysis to the non-convex setting in order to explain these phenomena [15, 18, 19, 20, 21, 22, 23, 24, 25]. However, none of these works proposed a unified framework able to handle both bounded and heavy-tailed noises.

This paper aims at filling this gap by providing a novel unified analysis of the convergence of SGD in a non-convex and noisy setting. With regards to the above mentioned works, our contribution is

threefold. First, we introduce a novel mathematical tool: biased expectation which allows to derive many results in stochastic analysis. Second, we show how to use this tool in the context of stochastic non-convex optimization to handle a large panel of noise assumptions. Third, the probabilistic bounds we obtain for SGD (i.e., on quantiles) provide novel insights over the previously known in-expectation bounds as they allow to consider heavy-tailed noise distributions with infinite variance, explaining why multi-start methods work by showing that a small number of runs of SGD will exhibit good convergence and not be disrupted by extreme noise.

The rest of the paper is organized as follows. In Section 2, we introduce the biased expectation and its main properties. In Section 3, we introduce the optimization framework of the analysis and present our main result. In Section 4, we explicit the convergence rates for various noise assumptions. Finally, the results obtained in the paper are illustrated in an empirical assessment in Section 5. All proofs can be found in the Supplementary Material provided as a separate document.

**Related work.**    Lower and upper bounds for first-order optimization in convex settings have been well-studied and understood in the literature (see, e.g., [11, 12, 3, 13]). Here, we focus on the results related to non-convex settings, and more specifically on the complexity of finding an $\varepsilon$-stationary point (i.e. a point $x_t$ such that $\mathbb{E}[\|\nabla f(x_t)\|^2] \leq \varepsilon$). First, several universal lower bounds have been provided for the convergence of any first-order algorithm [26, 27]. For smooth and noiseless setting, [26] established that $\Omega(\varepsilon^{-1})$ gradient evaluations are necessary for finding $\varepsilon$-stationary points; and showed that this rate is achieved by gradient descent. For smooth and bounded variance noise, [27] went on showing that $\Omega(\varepsilon^{-2})$ noisy gradient evaluations are required to reach an $\varepsilon$-stationary point, proving as a byproduct that the SGD is optimal with this worst case metric. With regards to the performance of SGD, [23] established an $O(\varepsilon^{-2})$ upper bound for the smooth, bounded variance and light-tail noise setting. Moreover, [28] went on showing that SGD itself cannot obtain a rate better than $\Omega(\varepsilon^{-2})$ in this noise setting, even for convex functions. For the smooth and heavy-tailed noise setting, [22] reports a complexity of $O(\varepsilon^{-b/(b-1)})$ where $b > 1$ denotes the tail-index for SGD using a slightly different Hölder-smoothness assumption. With regards to these works, our analysis allows to recover the standard results of SGD (Theorems 11 and 14) while extending the convergence rates to heavy-tailed noise (Theorem 17) using a single and unified analysis. In addition, we obtain novel results for biased noise (Theorem 12) as well as more generic bounds on quantiles instead of in-expectations (Theorems 14 and 12), explaining the convergence of multi-start strategies as well as the case of infinite variance noise. Finally, it is also worth mentioning that a recent line of works have been devoted to the design of algorithms that improve the convergence rate of SGD for non-convex problems using additional assumptions (we refer to [18] for a review). For instance, [29] used variance reduction techniques for the finite-sum problem allowing them to derive a convergence rate of order $O(\varepsilon^{-1})$ when all the component of the sum have bounded gradient and subsequent algorithms [30, 31, 32] have been provided to achieve a rate of order $O(\varepsilon^{-3/2})$.

**Notations.**    For all $x = (x_1, \ldots, x_d) \in \mathbb{R}^d$, we denote by $\|x\|^2 = \sum_{i=1}^d x_i^2$ the standard squared $\ell_2$-norm. A real-valued random variable $X$ is said to have a right (resp. left) heavy-tailed distribution if its moment generating function $\mathbb{E}\left[e^{sX}\right]$ is infinite for any $s > 0$ (resp. $s < 0$). We will denote as *heavy-tailed distribution* a distribution that is either right or left heavy-tailed. More precisely, we will say that $X$ follows a *fat-tailed* distribution of tail-index $\alpha > 0$ if $\mathbb{P}\left(|X| > x\right) = \Theta(x^{-\alpha})$. We will denote as *sub-exponential* a distribution that is not heavy-tailed (i.e., whose tail distribution is at most exponential), and will use the following standard definition: a real-valued random variable $X$ is $(\sigma^2, b)$-*sub-exponential* if, for all $|s| \leq 1/b$, $\ln \mathbb{E}\left[e^{s(X-\mu)}\right] \leq \frac{\sigma^2 s^2}{2}$, where $\mu = \mathbb{E}\left[X\right]$ and we recall that a $(\sigma^2, 0)$-sub-exponential random variable is also called $\sigma^2$-*sub-Gaussian*. Last, the notation a.s. stands for almost surely, r.v. for random variable and we respectively denote by $\mathcal{U}$, $\mathcal{N}$, $\mathcal{B}$ and $\mathcal{E}$ the standard uniform, normal, Bernoulli and exponential distributions.

## 2 Biased expectations: a simple operator to derive concentration inequalities

In this section, we introduce a simple yet effective mathematical tool to obtain concentration inequalities: *biased expectations*. This concept is an extension of the expectation of a random variable that allows for sharp concentration inequalities while retaining most of its mathematical properties. Its definition relies on *distortion functions* to bias the expectation: $\forall s, x \in \mathbb{R}$, let $\phi_s(x) = \frac{e^{sx}-1}{s}$ if $s \neq 0$, and $\phi_0(x) = x$ (see Figure 1). All distortion functions are non-decreasing, and $(s, x) \mapsto \phi_s(x)$

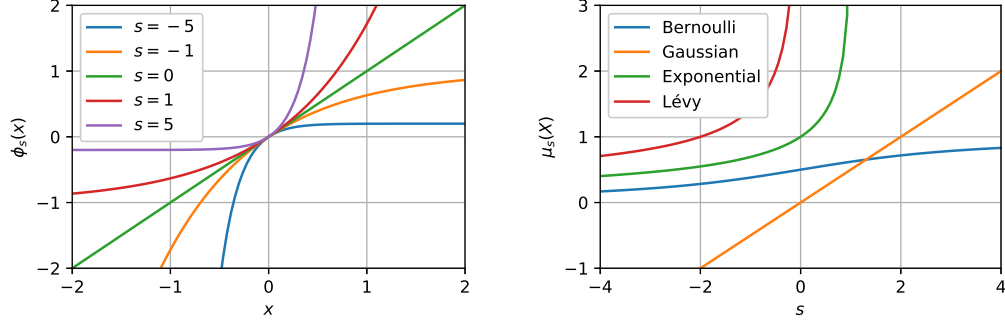

Figure 1: Distortion $\phi_s$ for different parameters.    Figure 2: Biased exp. for standard distributions.

is continuous. Moreover, $\phi_s$ matches the identity function for $s = 0$, and gradually bends to give more weight to higher (resp. lower) values for $s > 0$ (resp. $s < 0$). The intuition behind biased expectations is to distort the random variable $X$ using $\phi_s$ before taking its expectation.

**Definition 1** (Biased expectation). *Let $X$ be a real-valued random variable, and $I_X = \{s \in \mathbb{R} \mid \phi_s(X)$ is integrable$\}$. Then, for any $s \in I_X$, we denote as* biased expectation *of $X$ the quantity*

$$\mu_s(X) = \phi_s^{-1}\left(\mathbb{E}\left[\phi_s(X)\right]\right) . \tag{1}$$

First, note that the domain of definition $I_X$ is necessarily an interval, by application of dominated convergence, and $I_X$ contains $0$ if and only if $X$ is integrable. Moreover, a simple rewriting indicates that $\mu_s(X) = \frac{1}{s} \ln \mathbb{E}\left[e^{sX}\right]$. Hence, this quantity is tightly connected to the moment generating function $M_s(X) = \mathbb{E}\left[e^{sX}\right]$ and the cumulant generating function $\kappa_s(X) = \ln \mathbb{E}\left[e^{sX}\right]$. It is thus very natural for this quantity to allow for sharp concentration results on the underlying random variable $X$. Moreover, when $X$ is integrable, one can recover the expectation using $\mu_0(X) = \mathbb{E}\left[X\right]$. While other distortion types would be possible, we underline that the choice of the exponential for $\phi_s$ allows an important characteristic of the expectation to be preserved, as shown in the next results.

**Proposition 2** (Sum of independent variables). *Let $X$ and $Y$ be two independent real-valued random variables. Then, for any $s \in I_X \cap I_Y$, we have that $\mu_s(X + Y) = \mu_s(X) + \mu_s(Y)$.*

This proposition allows to separate a deterministic error from its random variation, as well as two independent sources of error. It is thus useful for proving convergence rates (see e.g. the proof of Theorem 1 in the Supplementary Material). Moreover, for positive values of $s > 0$, an alternative formulation of biased expectations using $L_p$-norms (i.e., $\|Y\|_p = \mathbb{E}\left[|Y|^p\right]^{1/p}$) provides insights into the properties of the operator: $\mu_s(X) = \ln \|e^X\|_s$. As a consequence, biased expectations interpolate between the minimum and the maximum of a random variable, as shown in the following result.

**Proposition 3.** *Let $X$ be a random variable and $\mu_s(X)$ its biased expectation. Then, the function $s \mapsto \mu_s(X)$ is continuous and non-decreasing on $I_X$, and*

   *1. If $I_X$ is left-unbounded, then $\lim_{s \to -\infty} \mu_s(X) = \operatorname{ess\,inf} X$,*

   *2. If $I_X$ is right-unbounded, then $\lim_{s \to +\infty} \mu_s(X) = \operatorname{ess\,sup} X$,*

   *3. If $X$ is integrable, then $\mu_0(X) = \mathbb{E}\left[X\right]$,*

*where $\operatorname{ess\,inf} X = \sup\{b \in \mathbb{R} : \mathbb{P}\left(X < b\right) = 0\}$ (resp. $\operatorname{ess\,sup} X = \inf\{b \in \mathbb{R} : \mathbb{P}\left(X > b\right) = 0\}$) is the essential infimum (resp. essential supremum) of $X$.*

Thus, biased expectations play a role similar to that of quantiles in the sense that they give an overall understanding of the whole probability distribution. As an example, Figure 2 shows the values of the biased expectations for four different random variables: Bernoulli ($X \sim \mathcal{B}(p)$), Gaussian ($X \sim \mathcal{N}(\mu, \sigma^2)$), exponential ($X \sim \mathcal{E}(\lambda)$) and Lévy ($X \sim \text{Lévy}(0, 1)$). Interestingly, Bernoulli r.v. have bounded biased expectations $\mu_s(X) \in [0, 1]$, Gaussian r.v. have linear biased expectations $\mu_s(X) = \mu + \frac{\sigma^2}{2} s$, exponential r.v. have positive biased expectations that diverge when $s \to \lambda^-$, and

for Lévy r.v., the biased expectations diverge when $s \to 0^-$ with rate $\mu_s(X) = \Theta(|s|^{-1/2})$. Before presenting our last characteristic, we further argue that conditioning with unbiased expectations is extremely easy, and the law of total expectation holds true.

**Proposition 4** (Law of total expectations)**.** *Let $X$ be a random variable and $\mathcal{F}$ be a sub-$\sigma$-algebra of the probability space of $X$. Then, denoting by $\mu_s(X|\mathcal{F}) = \phi_s^{-1} \ln \mathbb{E}\left[\phi_s(X)|\mathcal{F}\right]$ the biased expectation of $X$ conditioned by $\mathcal{F}$, the law of total expectations hold:*

$$\mu_s(\mu_s(X|\mathcal{F})) = \mu_s(X). \tag{2}$$

*Similarly, if $Y$ is a second random variable, then $\mu_s(\mu_s(X|Y)) = \mu_s(X)$.*

Finally, a key characteristic of biased expectations is that they are strongly linked to concentration inequalities, and upper bounding them will automatically imply bounds with high probability.

**Proposition 5** (Concentration inequalities)**.** *Let $X$ be a real-valued random variable. Then, for all $x, s > 0$ and $s \in I_X$, we have that*

$$\mathbb{P}\left(X \geq \mu_s(X) + x\right) \leq e^{-sx}. \tag{3}$$

*Moreover, if $X \geq 0$ almost surely, then $(-\infty, 0) \subset I_X$ and, for all $s > 0$,*

$$\mathbb{P}\left(X \geq x\right) \leq \frac{1 + sx}{x} \mu_{-s}(X). \tag{4}$$

While Eq. (3) is a rewriting of the Chernoff bound, Eq. (4) provides an improvement over the standard Markov inequality which can be rewritten as $\mathbb{P}\left(X \geq x\right) \leq \lim_{s \to 0^-} \mu_s(X)/x$. As we will see in the sequel, this improvement is particularly useful in the case of heavy-tailed distributions for which $\mathbb{E}\left[X\right]$ is infinite, but not $\mu_s(X)$ for a negative parameter $s < 0$ (see, e.g., the Lévy distribution in Figure 2).

The potential applications of this generic concept of biased expectations extend well beyond our current topic of interest, and a more detailed analysis of its mathematical properties and applications will be available in a follow-up paper. We now focus on its applications in stochastic optimization.

## 3 Theoretical analysis of stochastic gradient descent

In this section, we present the optimization framework considered in the analysis as well as the main result of the paper presented in its generic form over unbiased expectations.

**Setup.** Let $d \geq 1$ be an integer and $f : \mathbb{R}^d \to \mathbb{R}$ a real-valued function. We are interested in analyzing the following optimization problem

$$\min_{x \in \mathbb{R}^d} f(x) \tag{5}$$

where the objective function $f$ is assumed to $\beta$-smooth (i.e., its gradient is $\beta$-Lipschitz) and bounded from below, but possibly non-convex. More precisely, we focus on *stochastic gradient descent* (SGD), a simple yet efficient optimization algorithm widely used in the Machine Learning community to minimize the training loss, and more specifically for the training of neural networks. We consider here the standard noisy framework, where at each iteration, the optimization algorithm can access a noisy estimation $G_t = \nabla f(x_t) + X_t$ of the gradient of the objective function, where $X_t$ is the noise of the approximate gradient. The generic structure of the algorithm is displayed in Alg. 1. For instance, this optimization structure is of particular interest when the objective function $f(x) = \mathbb{E}\left[F(x, \xi)\right]$ is an expectation over a random variable $\xi$ representing individual samples of the dataset (e.g. images for object recognition) and an optimization algorithm can access the gradient $G_t = \nabla F(x, \xi_i)$ for a single data sample $\xi_i$, or a mini-batch estimation of the whole gradient $G_t = \frac{1}{m} \sum_{i=1}^{m} \nabla F(x, \xi_i)$ for some $m > 1$. In both cases, the partial gradient $G_t$ can be interpreted as an approximation of the true gradient of the objective $\nabla f(x_t)$ and the size of its noise $X_t = G_t - f(x_t)$ will impact the convergence rate of the algorithm. To derive an analysis of the SGD, we thus need to introduce our set of assumptions over the noise. Moreover, we will denote as $\Delta = f(x_0) - \min_{x \in \mathbb{R}^d} f(x)$ the difference between the initial and minimum function value in the remainder for clarity reasons.

Our main assumption uses biased expectations to bound the distance between the true gradient $\nabla f(x_t)$ and its approximate $G_t$ used at each iteration of Alg. 1. To account for biased as well as unbiased noise types, we decompose the error into two terms.

---

**Algorithm 1** Stochastic gradient descent (SGD)

---
**Input:** iterations $T$, gradient step $\eta$, initial state $x_0$
**Output:** optimizer $x_T$
 1: **for** $t = 0$ to $T - 1$ **do**
 2:     Compute $G_t$, the noisy approximation of $\nabla f(x_t)$
 3:     $x_{t+1} = x_t - \eta G_t$
 4: **end for**
 5: **return** $x_T$

---

**Assumption 6** (Biased variance). *Let $(\mathcal{F}_t)_{t \geq 0}$ be the filtration associated to the sequence of iterates $(x_t)_{t \geq 0}$ of Alg. 1. For all $s \in \mathbb{R}$, there exists $\sigma_s \in \mathbb{R}_+ \cup \{+\infty\}$ such that, for any $t \geq 0$,*

$$\mu_s \left( \|X_t\|^2 \mid \mathcal{F}_t \right) \leq \sigma_s^2 \,. \tag{6}$$

The variance term $\sigma_s^2$ directly accounts for the size of the noise. The second assumption we introduce is needed to account for its projection along the gradient, and ensure that most iterations of SGD move towards the minimum. Due to the added complexity of the heavy-tailed setting, we further decompose this term in two cases: positive and negative values of the parameter $s$.

**Assumption 7** (Biased mean). *Let $(\mathcal{F}_t)_{t \geq 0}$ be the filtration associated to the sequence of iterates $(x_t)_{t \geq 0}$ of Alg. 1. For all $s \geq 0$, there exists $m_s \in \mathbb{R}_+ \cup \{+\infty\}$ such that, for any $t \geq 0$,*

$$\mu_s \left( -\langle X_t, \nabla f(x_t) \rangle \mid \mathcal{F}_t \right) \leq m_s \,, \tag{7}$$

*and for all $s < 0$, there exists $m_s \in \mathbb{R}_+ \cup \{+\infty\}$ such that, for any $t \geq 0$,*

$$\frac{\mathbb{E}\left[ -\langle X_t, \nabla f(x_t) \rangle e^{s\|X_t\|^2} \mid \mathcal{F}_t \right]}{\mathbb{E}\left[ e^{s\|X_t\|^2} \mid \mathcal{F}_t \right]} \leq m_s \,. \tag{8}$$

Both terms in Assumption 7 ensure that approximate gradients are positively aligned with true gradients. While Eq. (7) biases the expectation towards high values of $-\langle X_t, \nabla f(x_t) \rangle$, Eq. (8) biases the expectation towards low values of $\|X_t\|^2$. Note also that, when the noise is integrable, both terms tend to $\mathbb{E}\left[ -\langle X_t, \nabla f(x_t) \rangle \mid \mathcal{F}_t \right]$ as $s \to 0$, and will thus tend to 0 for centered noise distributions. Equipped with these assumptions, we are now ready to cast our main result which relates the biased expectation of the minimum gradient norm to the biased variance and biased mean.

**Theorem 8** (Main result). *Fix any $\eta \in (0, 1/\beta]$ and consider that Assumption 6 and Assumption 7 hold true. Then, the biased expectation of the squared gradient norm averaged over $T$ iterations of Alg. 1 is upper-bounded as follows:*

$$\mu_s \left( \frac{1}{T} \sum_{t=0}^{T-1} \|\nabla f(x_t)\|^2 \right) \leq \frac{2\Delta}{\eta T} + 2(1 - \beta\eta)m_u + \beta\eta\sigma_v^2 \,, \tag{9}$$

*where $u = \frac{4s}{T}$ and $v = \frac{2\beta\eta s}{T}$ if $s \geq 0$, and $u = v = \frac{\beta\eta s}{T}$ otherwise.*

In other words, the distribution of the minimum gradient norm recorded after $T$ iterations is upper-bounded by the amount of noise on the gradient through its biased variance $\sigma_u^2$ and biased mean $m_v$. Note that, when $s = o(T)$, both parameters $u$ and $v$ tend to zero, and only the behavior of the biased expectations $\sigma_s^2$ and $m_s$ around $s = 0$ will play a role in the analysis. Moreover, when the noise is unbiased and integrable, we have $m_0 = 0$ and the second term in Eq. (9) will also tend to 0 allowing for the convergence of the minimum gradient norm. We will now investigate how these assumptions lead to concentration results for a large panel of noise distributions.

**Remark 9** (Varying step-size). Before presenting the next results, we stress that a direct generalization of the proof of Theorem 8 allows for varying gradient step $\eta_t$, as well as varying noise terms $m_{s,t}^2$ and $\sigma_{s,t}^2$. However, all the results presented below only consider fixed parameters for clarity purpose.

**Remark 10** (Bound on the minimum). Last, we point out that the results presented in the next sections remain valid on the minimum gradient norm, as $\min_{t < T} \|\nabla f(x_t)\|^2 \leq (1/T) \cdot \sum_{t=0}^{T-1} \|\nabla f(x_t)\|^2$.

# 4 Application to different noise types

In this section, we show how to derive convergence rates from Theorem 8 in several noise settings: unbiased with a bounded variance, deterministic, Gaussian and heavy-tailed.

## 4.1 Convergence in expectation for centered noise with a bounded variance

As discussed in Section 2, we have that $\mathbb{E}[X] = \mu_0(X)$ for any integrable random variable $X$. Hence, one can directly recover convergence rates for the expectation of the averaged gradient norm by taking $s = 0$ in Theorem 8. More precisely, considering unbiased noises ($m_0 = 0$) with finite variance ($\sigma_0 = \mathbb{E}\left[\|X_t\|^2 \mid \mathcal{F}_t\right]$), we obtain the following result for an optimal choice of the step $\eta$.

**Theorem 11.** *Assume that $\mathbb{E}[X_t|\mathcal{F}_t] = 0$ and $\mathrm{var}(X_t|\mathcal{F}_t) \leq \sigma^2$ for all $t < T$. Then, the expectation of the squared gradient norm averaged over $T$ iterations of Alg. 1 with $\eta = \min\left\{\sqrt{\frac{2\Delta}{T\beta\sigma^2}}, \frac{1}{\beta}\right\}$ is bounded by*

$$\mathbb{E}\left[\frac{1}{T}\sum_{t=0}^{T-1}\|\nabla f(x_t)\|^2\right] \leq \frac{4\beta\Delta}{T} + \sqrt{\frac{8\beta\Delta\sigma^2}{T}}. \tag{10}$$

This result shows that the convergence is of order $O(1/T)$ until the upper bound reaches the variance of the noise (or, more precisely, $4\sigma^2$), and then becomes of order $O(1/\sqrt{T})$. In other words, the number of iterations required to reach a precision $\varepsilon > 0$ on the expectation of the gradient norm is at most $O(\beta\Delta\sigma^2/\varepsilon^2)$. Interestingly, this convergence rate is of the same (optimal) order as the one reported in the previous works of [23, 24] in a similar non-convex setting with bounded variance noise. Quite surprisingly, we will also see in Section 4.4 that while this upper bound is infinite when the variance $\sigma^2$ is infinite, the gradient norm can still converge with high probability at the cost of a decrease in the convergence speed. Moreover, slightly biasing the expectation with $s > 0$ will also lead to convergence rates in high probability instead of in expectation, as indicated in Section 4.3.

## 4.2 Almost sure convergence for deterministic or bounded noise

As exhibited in Proposition 3, biased expectations also provide information on the maximum and minimum values taken by a random variable when $s \to \pm\infty$. As a consequence, analyzing Theorem 8 when $s \to +\infty$ allows to provide results for bounded or deterministic noises.

**Theorem 12.** *Assume that, $\forall t < T$, $\|G_t - \nabla f(x_t)\| \leq B$. Then, the squared gradient norm averaged over $T$ iterations of Alg. 1 with $\eta = 1/\beta$ is almost surely bounded by*

$$\frac{1}{T}\sum_{t=0}^{T-1}\|\nabla f(x_t)\|^2 \leq \frac{2\beta\Delta}{T} + B^2 \quad a.s.. \tag{11}$$

Thus, we recover a convergence of order $O(1/T)$ up to the (squared) error on the gradient $B^2$. The presence of the constant term $B^2$ comes from the fact that since the error can be deterministic (or even adversarial), the gradient norm cannot decrease further this threshold. Moreover, it is also interesting to note that as the noise tends to vanish (i.e. when $B \to 0$), then the algorithm is equivalent to a gradient descent (i.e. $G_t = \nabla f(x_t)$) and the upper bound provided in Theorem 3 matches the known convergence rate of order $O(1/T)$ of the gradient descent in non-convex settings (see, e.g., Theorem 1 in [25]).

## 4.3 Convergence with high probability for centered sub-exponential noise

Here, we investigate the implications of Theorem 8 for positive parameter $s > 0$. First, recall that an important aspect of Theorem 8 is that the biased expectations $\sigma_u^2$ and $m_v$ are considered for $u = \frac{2s\beta\eta}{T}$ and $v = \frac{4s}{T}$ that tend to zero as the number of iterations $T$ increases. As a consequence and keeping in mind that Proposition 5 provides sharp concentration inequalities when $s$ is positive, we will only need to analyze the behavior of biased expectations around $s = 0$ to derive convergence results with high probability. We thus formulate the following assumption on the noise to derive the next result.

**Assumption 13** (sub-exponential noise)**.** Let $(\mathcal{F}_t)_{t \geq 0}$ be the filtration associated to the sequence of iterates $(x_t)_{t \geq 0}$ of Alg. 1. We assume that the noise is centered (i.e., $\mathbb{E}\left[X_t \mid \mathcal{F}_t\right] = 0$) and its variance bounded by $\sigma^2$ (i.e., $\text{var}(X_t \mid \mathcal{F}_t) \leq \sigma^2$). Moreover, there exists $a, b, c > 0$ such that, conditional to $\mathcal{F}_t$, $\langle X_t, \nabla f(x_t) \rangle$ is $(a\sigma^2, c)$-sub-exponential, and $\|X_t\|^2$ is $\left(b\sigma^2, c\right)$-sub-exponential.

Under Assumption 13, the tail distribution of $\|X_t\|^2$ is at most exponential (since it is sub-exponential), and $\|X_t\|$ thus needs to be sub-Gaussian. Moreover, as the constant $a$ depends linearly on the norm of the gradient $\|\nabla f(x_t)\|$, we usually need the additional assumption that $f$ is $L$-Lipschitz to verify Assumption 13. As an example, Table 1 provides some constants $a, b, c$ that satisfy Assumption 13 for multiple standard distributions for an i.i.d. noise $X_t$ under the assumption that $f$ is $L$-Lipschitz. Moreover, we point out that, when the noise is centered and bounded by some constant $B$, a straightforward application of Hoeffding's Lemma [33] directly gives the values of the constants, allowing to cover a wide class of noise assumptions. We can now cast our result.

**Theorem 14.** *Assume that Assumption 13 is verified, let $\kappa_1 = 1 + b/2c$ be a constant. Then, with $\eta = \min\left\{\sqrt{\frac{2\Delta}{\kappa_1 \sigma^2 \beta T}}, \frac{1}{\beta}\right\}$, the squared gradient norm averaged over $T$ iterations of Alg. 1 is bounded, with probability at least $1 - \delta$, by*

$$\frac{1}{T} \sum_{t=0}^{T-1} \|\nabla f(x_t)\|^2 \leq \frac{4\beta\Delta}{T} + \frac{8c \ln(1/\delta)}{T} + \sqrt{\frac{8\kappa_1 \sigma^2 \beta \Delta}{T}} + \sqrt{\frac{16a\sigma^2 \ln(1/\delta)}{T}}. \qquad (12)$$

In other words, the number of iterations necessary to reach a precision $\varepsilon > 0$ is at most of order $O\left(\varepsilon^{-2} \ln(1/\delta)\right)$ with probability at least $1 - \delta$. With regards to existing work, it is to the best of our knowledge the first result that considers sub-exponential noise in a non-convex setting. However, it is also interesting to note that, similarly to Theorem 2, the rate we recover is still of order $O(\varepsilon^{-2})$. This might nonetheless not be surprising as the result also covers the sub-case of of bounded noise also covered in Theorem 2.

## 4.4 Convergence with high probability for centered heavy-tailed noise

Finally, we focus on heavy-tailed distributions with infinite variance which are difficult to handle using traditional mathematical tools. Fortunately, for any positive random variable $X \geq 0$, biased expectations are never infinite for negative parameter $s < 0$, since $\mathbb{E}\left[e^{sX}\right] \in (0, 1]$. Moreover, distributions whose expectation is infinite have biased expectations $\mu_s(X)$ that tend to $+\infty$ when $s \to 0^-$ (see, e.g., the Lévy distribution on Figure 2). As a consequence, bounding the biased expectation of the gradient norm when $s \to 0^-$ informs us about its heavy-tail behavior, and thus Theorem 8 can be used with $s < 0$ to obtain convergence rates with high probability. We may now introduce the corresponding noise assumption.

**Assumption 15.** Let $(\mathcal{F}_t)_{t \geq 0}$ be the filtration associated to the sequence of iterates $(x_t)_{t \geq 0}$ of Alg. 1. We assume that the noise is integrable and centered (i.e., $\mathbb{E}\left[X_t \mid \mathcal{F}_t\right] = 0$), and there exists constants $a, b, c > 0$ and $b \in (1, 2]$ such that, for all $s \in [0, 1/c]$,

$$\mu_{-s}\left(\|X_t\|^2 \mid \mathcal{F}_t\right) \leq a \cdot s^{\frac{b-2}{2}}. \qquad (13)$$

Note that $\mu_{-s}(X)$ near $s = 0$ is then tightly connected to the tail distribution of $X$. More specifically, Assumption 15 imposes that $\|X_t\|$ follows a fat-tailed distribution of tail index at least $b$.

**Proposition 16.** *If Assumption 15 is verified, then $\forall x \geq \sqrt{c}$, $\mathbb{P}\left(\|X_t\| \geq x \mid \mathcal{F}_t\right) \leq 2ax^{-b}$.*

Table 1: Examples of the constants satisfying Assumption 13 (assuming $f$ is $L$-Lipschitz) for different noise assumptions. All distributions were chosen so that $\text{var}(X_t \mid \mathcal{F}_t) \leq \sigma^2$.

| Distribution | $a$ | $b$ | $c$ | $\kappa_1 = 1 + \frac{b}{2c}$ |
|---|---|---|---|---|
| Centered Gaussian of covariance $\frac{\sigma^2}{d}I$ | $L^2/d$ | $8(2\ln(2) - 1)\sigma^2/d$ | $4\sigma^2/d$ | $2\ln(2)$ |
| Uniform on a ball of radius $\sigma$ | $L^2/4$ | $1/4$ | $1/8$ | $2$ |
| Uniform on a hypercube $[-\frac{\sigma}{\sqrt{d}}, \frac{\sigma}{\sqrt{d}}]^d$ | $L^2/4$ | $\sigma^2/2d$ | $\sigma^2/4d$ | $2$ |

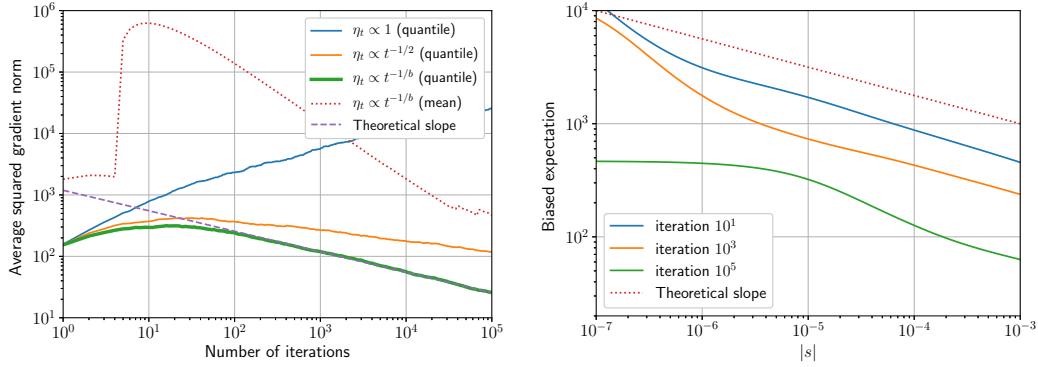

(a) Mean and 50%-quantiles of $\frac{1}{t}\sum_{i\leq t}\|\nabla f(x_i)\|^2$.      (b) Biased expectation $\mu_{-s}(\frac{1}{t}\sum_{i\leq t}\|\nabla f(x_i)\|^2)$.

Figure 3: SGD on a ridge regression problem with heavy-tail (Student's t) distribution of tail-index $b = 1.5$.

Additionally, in the case of symmetric noise (i.e., $X_t$ and $-X_t$ have the same distribution), the biased mean of Assumption 7 is equal to 0, as $\langle X_t, \nabla f(x_t)\rangle e^{s\|X_t\|^2}$ is antisymmetric w.r.t. $X_t$, and $\mathbb{E}[-\langle X_t, \nabla f(x_t)\rangle e^{s\|X_t\|^2} \mid \mathcal{F}_t] = 0$. The second case in which the biased mean is easy to bound is that of $L$-Lipschitz functions. Thus, using Assumption 15 to bound the biased variance directly leads to a convergence result in both cases.

**Theorem 17.** *Assume that $f$ is $L$-Lipschitz or that $X_t$ (conditional to $\mathcal{F}_t$) has a symmetric distribution. Then, if Assumption 15 is verified, there exists constants $(\kappa_i)_{i\geq 5}$ only depending on $a, b, c$ such that, with $\eta = \min\left\{\frac{T\varepsilon}{\beta}\left(\frac{4\beta\Delta}{abT^3\varepsilon^2}\right)^{\frac{2}{2+b}}, \frac{1}{\beta}\right\}$, the squared gradient norm averaged over $T$ iterations of Alg. 1 is bounded, with probability at least $1 - \delta$, by*

$$\frac{1}{T}\sum_{t=0}^{T-1}\|\nabla f(x_t)\|^2 \leq \frac{\kappa_2\beta\Delta}{T\delta} + \frac{\kappa_3\sqrt{\beta\Delta}}{T^{\frac{4-b}{4}}\delta^{\frac{2+b}{4}}} + \frac{\kappa_4 L^{\frac{2+b}{3b}}(\beta\Delta)^{\frac{b-1}{3b}}}{T^{\frac{b-1}{b}}\delta^{\frac{2+b}{3b}}} + \frac{\kappa_5\sqrt{\beta\Delta}}{T^{\frac{b-1}{b}}\delta^{\frac{2+b}{2b}}}, \qquad (14)$$

*where $\varepsilon$ is the right hand side of the above equation and the constants $\kappa_i$ are detailed in the proofs.*

Note that the last term of Eq. (14) has the worst dependency with regards to $T$ and $\delta$. Hence, the number of iterations required to reach a precision $\varepsilon$ is at most of order $O(\varepsilon^{-\frac{b}{b-1}}\delta^{-\frac{2+b}{2b-2}})$ with probability at least $1 - \delta$. This bound is thus of the exact same order as the one reported in the work of [22] where they consider a slightly different non-convex setting with heavy-tailed noise. Finally, it also has to be noted as $\varepsilon \propto T^{\frac{b-1}{b}}$ when $T$ is large enough, this result requires a step size $\eta \propto T^{-1/b}$, where $b$ is the tail-index, instead of the classic $\eta \propto T^{-1/2}$ for gradient noises with bounded variance.

## 5 Experiments

In this section, we illustrate the practical implications of the results obtained in the paper.

**Protocol.** The set of experiments consists in finding the parameters $x \in \mathbb{R}^d$ of a ridge regression that minimize the empirical penalized loss $f(x) = \|Y - \xi x\|^2 + \lambda\|x\|^2$ over the *Airfoil Self-Noise Data Set* taken from the UCI machine learning repository [34] denoted here by $(Y, \xi) \in \mathbb{R}^n \times \mathbb{R}^{d\times n}$ where $n = 1503$ and $d = 5$ and with a regularization parameter set to $\lambda = 10$. To illustrate our theoretical results (e.g. Theorem 17), we considered the noisy gradient approximation $G_t = \nabla f(x_t) + X_t$ where $X_t$ is a heavy-tail (Student's t) noise distribution of tail-index $b = 1, 5$. We considered three different step-size scenarii: (1) constant step-size $\eta_t = 10^{-4}$, (2) $\eta_t = 10^{-4} \cdot t^{-1/b}$ provided by Theorem 17 and (3) the standard $\eta_t = 10^{-4} \cdot t^{-1/2}$ traditionally used in SGD. For each step-size, we ran 1000 times Alg. 1 with a budget of $T = 10^5$ iterations starting from the solution of the non-penalized problem $x_0 = (\xi^T\xi)^{-1}\xi^T Y$. We then computed the empirical quantiles (with confidence

parameter set to $\delta = 50\%$), expectations and biased expectations of the series of the random variables $(1/t) \cdot \sum_{i=1}^{t} \|\nabla f(x_i)\|^2$ for each iteration $t = 1 \ldots T$.

**Results.** Results are displayed in Figure 3. These results show several aspects of the experiments: (1) the averaged expectation $\frac{1}{t} \sum_{i=1}^{t} \|\nabla f(x_i)\|^2$ reaches extremely large values (infinite in theory) compared to the values of quantiles. (2) The choice of the step-size $\eta_t \propto t^{-1/b}$ does lead to the convergence rate of order $t^{(b-1)/b}$ exhibited in Theorem 17 (see the comparison with the theoretical bound displayed in Figure 3). (3) The standard step-sizes $\eta_t \propto t^{-1/2}$ and $\eta_t \propto 1$ (independent of the desired precision $\varepsilon$) lead to suboptimal convergence rates, indicating that the choice $\eta_t \propto t^{-1/b}$ may be valuable for practitioners when the noise distribution is particularly fat-tailed. (4) Biased expectations $\mu_{-s}(1/t \sum_{i=1}^{t} \|\nabla f(x_i)\|^2)$ are well aligned with the results of order $s^{(b-1)/2}$ stated in Equation 14.

## 6    Conclusion

This paper proposed a novel unifying analysis of stochastic gradient descent in the noisy and non-convex setting. We introduced a novel operator: unbiased expectations that provide powerful tools for stochastic analysis. Using this tool, we showed that SGD is robust in the non-convex setting over a large panel of noise assumptions, including infinite variance heavy-tailed noises.

## Broader impact

Based of the theoretical nature of the work, the authors do not believe this section is applicable to the present contribution, as its first goal is to provide some insights on a classical algorithm of the machine learning community and does not provide novel applications per se.

## Acknowledgments and Disclosure of Funding

The authors thank the whole team at Huawei Paris for useful discussions. This work was financed by Huawei Technologies.

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
