[Supplementary Material]

# Supplementary Material

This document contains the proofs of the results presented in the paper: Robustness Analysis of Non-Convex Stochastic Gradient Descent using Biased Expectations.

**Proof of Proposition 2.** If $s = 0$, the result is trivial. Otherwise, we have $\mu_s(X + Y) = \frac{1}{s} \ln \mathbb{E}\left[e^{s(X+Y)}\right] = \frac{1}{s} \ln\left(\mathbb{E}\left[e^{sX}\right]\mathbb{E}\left[e^{sY}\right]\right) = \mu_s(X) + \mu_s(Y)$, where the second equality follows from the independence of $e^{sX}$ and $e^{sY}$.

**Proof of Proposition 3.** First, note that $\mu_s(X) = -\mu_{-s}(-X)$, and thus 2 implies 1 (as $-\operatorname{ess\,sup}(-X) = \operatorname{ess\,inf} X$). Moreover, $L_p$-norms of probability spaces are both non-decreasing and tend to the essential supremum (i.e., $p \mapsto \|Y\|_p$ is non-decreasing and $\lim_{p \to +\infty} \|Y\|_p = \operatorname{ess\,sup} Y$). Hence, using the alternative formulation $\mu_s(X) = \ln \|e^X\|_s$, we get that $s \mapsto \mu_s(X)$ is non-decreasing, and $\lim_{s \to +\infty} \mu_s(X) = \ln(\operatorname{ess\,sup}(e^X)) = \operatorname{ess\,sup} X$. Finally, note that the function $(s, x) \mapsto \phi_s(x) = \frac{e^{sx}-1}{s}$ is continuous. Let $s_0, s_1 \in I_X$, and $s_0 < s < s_1$. By definition of $I_X$, $\phi_{s_0}(X)$ and $\phi_{s_1}(X)$ are integrable. Moreover, $|\phi_s(X)| = \max\{-\phi_s(X), \phi_s(X)\} \le \max\{-\phi_{s_0}(X), \phi_{s_1}(X)\} \le -\phi_{s_0}(X) + \phi_{s_1}(X) \le |\phi_{s_0}(X)| + |\phi_{s_1}(X)|$ by monotonicity of $s \mapsto \phi_s(x)$. As $|\phi_{s_0}(X)| + |\phi_{s_1}(X)|$ is integrable and independent of $s$, dominated convergence implies continuity of $\mathbb{E}\left[\phi_s(X)\right]$, and thus of $\mu_s(X)$, in $(s_1, s_2)$.

**Proof of Proposition 4.** A simple rewriting of $\mu_s(\mu_s(X|\mathcal{F}))$ leads to the desired result: $\mu_s(\mu_s(X|\mathcal{F})) = \phi_s^{-1}\left(\mathbb{E}\left[\phi_s \circ \phi_s^{-1}\left(\mathbb{E}\left[\phi_s(X)|\mathcal{F}\right]\right)\right]\right) = \phi_s^{-1}\left(\mathbb{E}\left[\mathbb{E}\left[\phi_s(X)|\mathcal{F}\right]\right]\right) = \phi_s^{-1}\left(\mathbb{E}\left[\phi_s(X)\right]\right) = \mu_s(X)$.

**Proof of Proposition 5.** Eq. (3) follows from the Chernoff bound $\mathbb{P}(X \ge a) \le \mathbb{E}\left[e^{sX}\right]e^{-sa}$ for $a = \mu_s(X) + x$. Moreover, if $X \ge 0$ a.s., using Markov's inequality on $\phi_s(X) \ge 0$ a.s. gives, $\forall x > 0$,

$$\mathbb{P}(X \ge x) \le \frac{\phi_s\left(\mu_s(X)\right)}{\phi_s(x)} . \tag{15}$$

When $s < 0$, we can further simplify Eq. (15) by using $\phi_s(\mu_s(X)) \le \mu_s(X)$ (as $\phi_s$ is concave), and $\phi_s(x) \ge \frac{x}{1-sx}$, which concludes the proof.

**Proof of Theorem 8.** The result follows from standard analysis of non-convex gradient descent. More specifically, using the $\beta$-smoothness of $f$, we have

$$\begin{aligned}
f(x_{t+1}) &\le f(x_t) + \langle \nabla f(x_t), x_{t+1} - x_t \rangle + \frac{\beta}{2}\|x_{t+1} - x_t\|^2 \\
&\le f(x_t) - \eta\langle \nabla f(x_t), G_t \rangle + \frac{\beta\eta^2}{2}\|G_t\|^2 \\
&\le f(x_t) - \eta\|\nabla f(x_t)\|^2 - \eta\langle \nabla f(x_t), X_t \rangle + \frac{\beta\eta^2}{2}\|\nabla f(x_t) + X_t\|^2 \\
&\le f(x_t) - \eta\left(1 - \frac{\beta\eta}{2}\right)\|\nabla f(x_t)\|^2 - \eta(1 - \beta\eta)\langle \nabla f(x_t), X_t \rangle + \frac{\beta\eta^2}{2}\|X_t\|^2
\end{aligned} \tag{16}$$

Rearranging Eq. (16) and summing over all times $t \in \{0, T-1\}$ leads to

$$\eta\left(1 - \frac{\beta\eta}{2}\right)\sum_{t<T}\|\nabla f(x_t)\|^2 \le \Delta - \eta(1 - \beta\eta)\sum_{t<T}\langle X_t, \nabla f(x_t)\rangle + \frac{\beta\eta^2}{2}\sum_{t<T}\|X_t\|^2, \tag{17}$$

where $\Delta = f(x_0) - \min_{x \in \mathbb{R}^d} f(x)$. Finally, using the assumption $\eta \in (0, 1/\beta]$ we obtain $\frac{\eta}{2} \le \eta\left(1 - \frac{\beta\eta}{2}\right)$, and thus dividing by $\eta T/2$ gives that

$$\frac{1}{T}\sum_{t=1...T}\|\nabla f(x_t)\|^2 \le \frac{2\Delta}{\eta T} - \frac{2(1-\beta\eta)}{T}\sum_{t<T}\langle X_t, \nabla f(x_t)\rangle + \frac{\beta\eta}{T}\sum_{t<T}\|X_t\|^2 . \tag{18}$$

To conclude, we apply biased expectation to both sides of Eq. (18). As $\langle X_t, \nabla f(x_t)\rangle$ and $\|X_t\|^2$ are not independent, Proposition 2 does not apply. We thus use the following Lemma to decompose the error.

**Lemma 18.** *Let $X, Y$ be two (possibly dependent) random variables and $s \in \mathbb{R}$. If $s \geq 0$, then $\mu_s(X + Y) \leq \mu_{2s}(X) + \mu_{2s}(Y)$. Otherwise, $\mu_s(X + Y) \leq \mu_s(X) + \frac{\mathbb{E}[Ye^{sX}]}{\mathbb{E}[e^{sX}]}$, whenever the right-hand sides are well-defined.*

*Proof.* For $s > 0$, applying the Cauchy-Schwartz inequality to $e^{sX}$ and $e^{sY}$ gives $\mu_s(X + Y) = \frac{1}{s} \ln \left( \mathbb{E}\left[e^{sX}e^{sY}\right] \right) \leq \frac{1}{2s} \ln \left( \mathbb{E}\left[e^{2sX}\right] \mathbb{E}\left[e^{2sY}\right] \right) = \mu_{2s}(X) + \mu_{2s}(Y)$. For $s < 0$, we obtain that $\mu_s(X + Y) = \mu_s(X) + \frac{1}{s} \ln \mathbb{E}\left[e^{sY} \frac{e^{sX}}{\mathbb{E}[e^{sX}]}\right]$ by a direct rewriting. Now, introducing the random variable $Y'$ with density $\frac{e^{sx}}{\mathbb{E}[e^{sX}]}$ w.r.t. the probability measure of $(X, Y)$ and using Jensen's inequality on the function $x \mapsto \frac{1}{s} \ln(x)$, we obtain that $\frac{1}{s} \ln \mathbb{E}\left[e^{sY} \frac{e^{sX}}{\mathbb{E}[e^{sX}]}\right] = \frac{1}{s} \ln \mathbb{E}\left[e^{sY'}\right] \leq \mathbb{E}\left[\frac{1}{s} \ln e^{sY'}\right] = \mathbb{E}\left[Y'\right] = \mathbb{E}\left[Y \frac{e^{sX}}{\mathbb{E}[e^{sX}]}\right]$ which proves the result.

$\square$

Moreover, note that, for any $a \in \mathbb{R}$, $\mu_s(aX) = a\mu_{as}(X)$. Then, using Proposition 2 to remove the deterministic error, we have

$$
\begin{aligned}
\mu_s \left( \frac{1}{T} \sum_{t=1...T} \|\nabla f(x_t)\|^2 \right) &\leq \mu_s \left( \frac{2\Delta}{\eta T} - \frac{2(1-\beta\eta)}{T} \sum_{t<T} \langle X_t, \nabla f(x_t) \rangle + \frac{\beta\eta}{T} \sum_{t<T} \|X_t\|^2 \right) \\
&\leq \frac{2\Delta}{\eta T} + \mu_s \left( \sum_{t<T} A_t \right),
\end{aligned}
$$
(19)

where $A_t = -\frac{2(1-\beta\eta)}{T} \langle X_t, \nabla f(x_t) \rangle + \frac{\beta\eta}{T} \|X_t\|^2$. Using Lemma 18 with $X = \frac{\beta\eta}{T} \|X_t\|^2$ and $Y = -\frac{2(1-\beta\eta)}{T} \langle X_t, \nabla f(x_t) \rangle$, we have $\mu_s(A_t \mid \mathcal{F}_t) \leq \frac{2(1-\beta\eta)}{T} m_u + \frac{\beta\eta}{T} \sigma_v^2$, where $u, v$ are defined as in Theorem 8 and $m_s, \sigma_s^2$ as in Assumption 6 and Assumption 7. Finally, we use Proposition 4 to bound the sums over iterations:

$$
\begin{aligned}
\mu_s \left( \sum_{t<T} A_t \right) &= \mu_s \left( \mu_s \left( \sum_{t<T} A_t \mid \mathcal{F}_{T-1} \right) \right) \\
&= \mu_s \left( \sum_{t<T-1} A_t + \mu_s \left( A_{T-1} \mid \mathcal{F}_{T-1} \right) \right) \\
&\leq \mu_s \left( \sum_{t<T-1} A_t + \frac{2(1-\beta\eta)}{T} m_u + \frac{\beta\eta}{T} \sigma_v^2 \right) \\
&= \mu_s \left( \sum_{t<T-1} A_t \right) + \frac{2(1-\beta\eta)}{T} m_u + \frac{\beta\eta}{T} \sigma_v^2 \\
&\leq 2(1-\beta\eta) m_u + \beta\eta\sigma_v^2,
\end{aligned}
$$
(20)

which concludes the proof.

In order to simplify our convergence rates, we will use the following lemma.

**Lemma 19.** *Let $a, b, c, p > 0$ and $f(x) = ax^p + b/x$. Then, with $x^* = \min\left\{ \left(\frac{b}{pa}\right)^{\frac{1}{1+p}}, c \right\}$, we have*

$$
f(x^*) \leq (1 + p^{-1})bc^{-1} + (1 + p)p^{\frac{-p}{1+p}} a^{\frac{1}{1+p}} b^{\frac{p}{1+p}}.
$$
(21)

*Proof.* When $b < pac^{1+p}$, we have $x^* = \left(\frac{b}{pa}\right)^{\frac{1}{1+p}}$ and $f(x^*) = \left(\frac{b}{pa}\right)^{\frac{1}{1+p}}$. Otherwise, we have $x^* = c$ and $f(x^*) = ac^p + b/c \leq (1 + p^{-1})b/c$. Hence, $f(x^*)$ is inferior to the sum of both terms. $\square$

**Proof of Theorem 11.** First, note that all the r.v. are integrable since the variance of the noise is bounded. Hence, for all the considered r.v. $X$, we have $\mu_0(X) = \mathbb{E}[X]$ (see Proposition 3), and Theorem 8 gives us that $\mathbb{E}\left[(1/T) \cdot \sum_{t=1}^{T} \|\nabla f(x_t)\|^2\right] \leq \frac{2\Delta}{\eta T} + \beta\eta\sigma^2$ when $s = 0$. Minimizing the right-hand side term over $\eta \in (0, 1/\beta]$ using Lemma 19 leads to the desired result.

**Proof of Theorem 12.** Using Proposition 3, we have $\lim_{s \to +\infty} \mu_s\left(\|X_t\|^2 \mid \mathcal{F}_t\right) = \operatorname{ess\,sup} \|X_t\|^2 \leq B^2$. Theorem 8 with $s \to +\infty$ and $\eta = 1/\beta$ thus gives $\operatorname{ess\,sup}\left((1/T) \sum_{t=1}^{T} \|\nabla f(x_t)\|^2\right) \leq \frac{2\beta\Delta}{T} + B^2$.

**Proof of Theorem 14.**    By definition of sub-exponential r.v., we have, $\forall s \in (0, 1/c]$, $\mu_s(-\langle X_t, \nabla f(x_t) \rangle \mid \mathcal{F}_t) \leq a\sigma^2 s/2$ and $\mu_s(\|X_t\|^2 \mid \mathcal{F}_t) \leq (1 + b/2c)\sigma^2$. Using Proposition 5 and Theorem 8 we thus have, $\forall x, s > 0$ such that $u = \frac{4s}{T} \leq 1/c$ and $v = \frac{2\beta\eta s}{T} \leq 1/c$,

$$\mathbb{P}\left(\frac{1}{T}\sum_{t=0}^{T-1}\|\nabla f(x_t)\|^2 \geq \frac{2\Delta}{\eta T} + 2(1-\beta\eta)m_u + \beta\eta\sigma_v^2 + x\right) \leq e^{-sx}, \tag{22}$$

where $m_u = a\sigma^2 u/2$ and $\sigma_v^2 = (1 + b/2c)\sigma^2$. Hence, if $\eta \in (0, 1/\beta]$, we have, with probability at least $1 - \delta$,

$$\frac{1}{T}\sum_{t=1}^{T}\|\nabla f(x_t)\|^2 \leq \frac{2\Delta}{\eta T} + \frac{4a\sigma^2 s}{T} + (1 + b/2c)\beta\eta\sigma^2 + \frac{1}{s}\ln(1/\delta). \tag{23}$$

Optimizing over $\eta$ and $s$ gives $\beta\eta = \min\left\{\sqrt{\frac{2\beta\Delta}{(1+b/2c)T\sigma^2}}, 1\right\}$ and $\frac{4cs}{T} = \min\left\{\sqrt{\frac{4c^2\ln(1/\delta)}{a\sigma^2 T}}, 1\right\}$. Using Lemma 19, Eq. (23) thus becomes

$$\frac{1}{T}\sum_{t=1}^{T}\|\nabla f(x_t)\|^2 \leq \frac{4\beta\Delta + 8c\ln(1/\delta)}{T} + \sqrt{\frac{8(1+b/2c)\beta\Delta\sigma^2}{T}} + 4\sigma\sqrt{\frac{a\ln(1/\delta)}{T}}. \tag{24}$$

**Proof of Proposition 16.** Using the second concentration inequality of Proposition 5, we have, $\forall x \geq \sqrt{c}$,

$$\mathbb{P}\left(\|X_t\| \geq x \mid \mathcal{F}_t\right) = \mathbb{P}\left(\|X_t\|^2 \geq x^2 \mid \mathcal{F}_t\right) \leq \frac{2\mu_{-1/x^2}(\|X_t\|^2 \mid \mathcal{F}_t)}{x^2} \leq 2ax^{-b}. \tag{25}$$

**Proof of Theorem 17.** We first bound the biased mean in both settings. If the noise is symmetric, then $\mathbb{E}\left[-\langle X_t, \nabla f(x_t) \rangle e^{s\|X_t\|^2} \mid \mathcal{F}_t\right] = 0$ and, for $s > 0$, $m_{-s} = 0$ verifies Assumption 7. Otherwise, we use the following Lemma.

**Lemma 20.** *If $f$ is L-Lipschitz and Assumption 15 is verified, then, $\forall s \in [0, 1/c]$,*

$$\frac{\mathbb{E}\left[-\langle X_t, \nabla f(x_t) \rangle e^{-s\|X_t\|^2} \mid \mathcal{F}_t\right]}{\mathbb{E}\left[e^{-s\|X_t\|^2} \mid \mathcal{F}_t\right]} \leq \kappa_6 L s^{\frac{b-1}{2}}, \tag{26}$$

*where $\kappa_6 = (1 - ac^{-b/2})^{-1}\left(c^{\frac{b}{2}} + \frac{4ab}{(b-1)(3-b)}\right)$.*

*Proof.* First, we have

$$\begin{aligned}
\mathbb{E}\left[e^{-s\|X_t\|^2} \mid \mathcal{F}_t\right] &= e^{-s\mu_{-s}(\|X_t\|^2 \mid \mathcal{F}_t)} \\
&\geq 1 - s\mu_{-s}(\|X_t\|^2 \mid \mathcal{F}_t) \\
&\geq 1 - as^{b/2} \\
&\geq 1 - ac^{-b/2}.
\end{aligned} \tag{27}$$

Then, let $Y = -\langle X_t, \nabla f(x_t) \rangle$. As $\mathbb{E}[Y \mid \mathcal{F}_t] = 0$, we have

$$\begin{aligned}
\mathbb{E}\left[Ye^{-s\|X_t\|^2} \mid \mathcal{F}_t\right] &= \mathbb{E}\left[Y_+ e^{-s\|X_t\|^2} \mid \mathcal{F}_t\right] - \mathbb{E}\left[Y_- e^{-s\|X_t\|^2} \mid \mathcal{F}_t\right] \\
&\leq \mathbb{E}[Y_+ \mid \mathcal{F}_t] - \mathbb{E}\left[Y_- e^{-s\|X_t\|^2} \mid \mathcal{F}_t\right] \\
&= \mathbb{E}\left[Y_-\left(1 - e^{-s\|X_t\|^2}\right) \mid \mathcal{F}_t\right] \\
&\leq L\mathbb{E}\left[\|X_t\|\left(1 - e^{-s\|X_t\|^2}\right) \mid \mathcal{F}_t\right],
\end{aligned} \tag{28}$$

as $Y_- \le |\langle X_t, \nabla f(x_t)\rangle| \le L\|X_t\|$. Finally, we bound $\mathbb{E}\left[\|X_t\|\left(1 - e^{-s\|X_t\|^2}\right) \mid \mathcal{F}_t\right]$ by using the function $g(x) = x\left(1 - e^{-sx^2}\right)$. As $g$ is monotonically increasing, we have

$$
\begin{aligned}
\mathbb{E}\left[\|X_t\|\left(1 - e^{-s\|X_t\|^2}\right) \mid \mathcal{F}_t\right] &= \mathbb{E}\left[g\left(\|X_t\|\right)\right] \\
&= \int_0^{+\infty} \mathbb{P}\left(g(X) > x\right) dx \\
&= \int_0^{+\infty} \mathbb{P}\left(X > g^{-1}(x)\right) dx \\
&= \int_0^{g(\sqrt{c})} \mathbb{P}\left(X > g^{-1}(x)\right) dx + \int_{g(\sqrt{c})}^{+\infty} \mathbb{P}\left(X > g^{-1}(x)\right) dx \\
&\le g(\sqrt{c}) + 2a \int_0^{+\infty} \left(g^{-1}(x)\right)^{-b} dx \\
&\le sc^{3/2} + 2a \int_0^{+\infty} \min\left\{x, (x/s)^{1/3}\right\}^{-b} dx \\
&\le sc^{3/2} + \frac{4ab}{(b-1)(3-b)} s^{\frac{b-1}{2}} \\
&\le \left(c^{\frac{b}{2}} + \frac{4ab}{(b-1)(3-b)}\right) s^{\frac{b-1}{2}},
\end{aligned}
$$

(29)

where the second inequality comes from $g(x) \le sx^3$ and $g(x) \ge \min\left\{x, (x/s)^{1/3}\right\}$, and the last inequality from $s \le 1/c$. $\qquad\square$

Hence, we can use $m_{-s} = \kappa_6 L s^{\frac{b-1}{2}}$, with the special case $L = 0$ if $X_t$ is symmetric, in order to describe both settings. Using Theorem 8 and Assumption 15, we thus have, for $s \in \left[0, \frac{T}{\beta\eta c}\right]$,

$$
\mu_{-s}\left(\frac{1}{T}\sum_{t=1}^T \|\nabla f(x_t)\|^2\right) \le \frac{2\Delta}{\eta T} + \beta\eta a\left(\frac{\beta\eta s}{T}\right)^{\frac{b-2}{2}} + \kappa_6 L\left(\frac{\beta\eta s}{T}\right)^{\frac{b-1}{2}}. \tag{30}
$$

We obtain a concentration inequality using Proposition 5, leading to, $\forall x > 0$,

$$
\mathbb{P}\left(\frac{1}{T}\sum_{t=1}^T \|\nabla f(x_t)\|^2 \ge x\right) \le \frac{1+sx}{x}\left[\frac{2\Delta}{\eta T} + \beta\eta a\left(\frac{\beta\eta s}{T}\right)^{\frac{b-2}{2}} + \kappa_6 L\left(\frac{\beta\eta s}{T}\right)^{\frac{b-1}{2}}\right]. \tag{31}
$$

Choosing $s = \min\left\{\frac{1}{x}, \frac{T}{\beta\eta c}\right\}$ gives $\frac{1+sx}{x} \le \frac{2}{x}$, $\left(\frac{\beta\eta s}{T}\right)^{\frac{b-1}{2}} \le \left(\frac{\beta\eta}{Tx}\right)^{\frac{b-1}{2}}$ (as $b \ge 1$) and $\left(\frac{\beta\eta s}{T}\right)^{\frac{b-2}{2}} \le \left(\frac{\beta\eta}{Tx}\right)^{\frac{b-2}{2}} + c^{\frac{2-b}{2}}$. Hence, with $y = \frac{\beta\eta}{Tx}$, we have

$$
\mathbb{P}\left(\frac{1}{T}\sum_{t=1}^T \|\nabla f(x_t)\|^2 \ge x\right) \le \frac{2}{x}\left[\frac{2\beta\Delta}{T^2 xy} + aTxy^{\frac{b}{2}} + ac^{\frac{2-b}{2}}Txy + \kappa_6 Ly^{\frac{b-1}{2}}\right], \tag{32}
$$

and optimizing the first two terms over $y$ (and thus $\eta$) gives $y = \min\left\{\left(\frac{4\beta\Delta}{abT^3 x^2}\right)^{\frac{2}{2+b}}, \frac{1}{Tx}\right\}$. Lemma 19 then gives

$$
\mathbb{P}\left(\frac{1}{T}\sum_{t=1}^T \|\nabla f(x_t)\|^2 \ge x\right) \le A + B + C + D, \tag{33}
$$

where

$$
\begin{aligned}
A &= \frac{4(2+b)\beta\Delta}{bTx} \\
B &= (2+b)\left(\frac{b}{2}\right)^{\frac{-b}{2+b}} x^{-1}(aTx)^{\frac{2}{2+b}}\left(\frac{2\beta\Delta}{T^2 x}\right)^{\frac{b}{2+b}} \\
C &= 2ac^{\frac{2-b}{2}}T\left(\frac{4\beta\Delta}{abT^3 x^2}\right)^{\frac{2}{2+b}} \\
D &= \frac{2\kappa_6 L}{x}\left(\frac{4\beta\Delta}{abT^3 x^2}\right)^{\frac{b-1}{2+b}}.
\end{aligned}
$$

(34)

Using $A + B + C + D \le \max\{4A, 4B, 4C, 4D\}$ and bounding the previous term by $\delta$, we get, with probability $1 - \delta$,

$$
\frac{1}{T}\sum_{t=1}^T \|\nabla f(x_t)\|^2 \le \frac{\kappa_2 \beta\Delta}{T\delta} + \frac{\kappa_3\sqrt{\beta\Delta}}{T^{\frac{4-b}{4}}\delta^{\frac{2+b}{4}}} + \frac{\kappa_4 L^{\frac{2+b}{3b}}(\beta\Delta)^{\frac{b-1}{3b}}}{T^{\frac{b-1}{b}}\delta^{\frac{2+b}{3b}}} + \frac{\kappa_5\sqrt{\beta\Delta}}{T^{\frac{b-1}{b}}\delta^{\frac{2+b}{2b}}}, \tag{35}
$$

where

$$
\begin{aligned}
\kappa_2 &= 16(2+b)/b & &\leq & &36 \\
\kappa_3 &= 2 \cdot 8^{\frac{2+b}{4}} c^{\frac{4-b^2}{8}} b^{-\frac{1}{2}} a^{\frac{b}{4}} & &\leq & &12 c^{\frac{4-b^2}{8}} a^{\frac{b}{4}} \\
\kappa_4 &= 2^{\frac{5b+4}{3b}} \kappa_6^{\frac{2+b}{3b}} (ab)^{\frac{1-b}{3b}} & &\leq & &8 \kappa_6^{\frac{2+b}{3b}} a^{\frac{1-b}{3b}} \\
\kappa_5 &= 2 \cdot (4 \cdot (2+b))^{\frac{2+b}{2b}} b^{-1/2} a^{1/b} & &\leq & &84 a^{1/b} \\
\kappa_6 &= (1 - ac^{-b/2})^{-1} \left( c^{\frac{b}{2}} + \frac{4ab}{(b-1)(3-b)} \right) & &\leq & &(1 - ac^{-b/2})^{-1} \left( c^{\frac{b}{2}} + \frac{8a}{b-1} \right)
\end{aligned}
\tag{36}
$$