[Reviews · NeurIPS 2020]

Review 1

Summary and Contributions: This paper analyzes stochastic gradient descent for smooth non-convex objectives with potentially heavy-tailed noise in the gradient oracle. A new method of quantifying tails, the “biased expectation” is introduced, that allows the others to generate high-probability bounds for quantities involving heavy tailed random variables. The analysis recovers standard results for subgaussian and possibly biased noise in the gradient, but also allows for proving convergence for certain classes of noise that have polynomially decaying tails.

Strengths: I liked the biased expectation trick, which seems like a very natural way to analyze the problem. The results provide concrete scalings for the learning rate parameter. Once the biased expectation is defined ,the results seem relatively straightforward applications of the definition. I do not think this is a disadvantage since easier proofs tend to be more generally useful in my opinion.

Weaknesses: One shortcoming I see is that the result holds only for some unknown iterate along the optimization path, rather than e.g. the last iterate. However, this is hard to show even for ordinary light-tailed noise distributions. I would also have liked to see more discussion of specific kinds of heavy tailed noise and their biased expectation values. ---- I have read the reviews and responses. I still think this is a good paper, although there are a few presentation issues. In line 413, I think the statement e^(sX)/E[e^(sX)] sums to one should be replaced with something like p(X,Y)e^(sX)/E[e^(sX)]. Also, please use a single counter for all theorems/lemmas/propositions. It makes it easier to binary search since any given page is likely to have at least one of these.

Correctness: I believe so, but I did not check all details in the appendix.

Clarity: yes

Relation to Prior Work: yes, to the best of my knowledge.

Reproducibility: Yes

Additional Feedback:


Review 2

Summary and Contributions: The paper presents a new framework (of 'biased expectations') to give an analysis of convergence of the plain SGD algorithm for non-convex but Lipschitz-gradient objective functions under various assumptions on the noise of the gradient estimator.

Strengths: The 'biased expectations' seems to be very closely related to log-moment generating functions as used extensively in applied probability. The authors provide claims of varying analysis under different noise assumptions.

Weaknesses: Post-feedback: Thanks for the detailed responses, they are satisfactory to me. --------- The analysis of convergence of SGD is a well-beaten track. To be able to get a new piece of work in this field, one needs to very clearly lay out the limitations of prior work that your work is able to remedy. Most of the SGD analysis concentrates on convergence-in-expectation, for which simple assumptions on bounds on bias in and variance of gradient estimation suffice to show convergence. For non-convex functions, convergence is in terms of either the average, weighed average or min of the norms of the gradients at each iterate. (The authors use the min of the norms.) However, I do not immediately get a sense on why this new framework and analysis is needed. Why should I care about convergence in (high) probability over in-expectation convergence? I understand that the characteristics of the bounds vary by the fatness of the tail of the gradient's noise because in-probability convergence now needs to worry about the rarer cases of slower convergence. But to get work on this accepted, there has to be a lot more effort, e.g. using simple concrete example objective functions and resultant iterations, or via extensive numerical experiments to show that in-probability convergence matters. There is a glimmer of this in Fig 3, but the message is muddied by the fact that both bounded but uncentered noise and heavy tailed distributions do worse. Clearly a bias-variance trade-off is happening here and so it is not just the tail of the noise that matters, but the theoretical results seem silent on this.

Correctness: I have not verified the proofs of the claims, but they seem to align correctly with what I would expect. One curious or worrisome factor seems to be their dependence on the optimality gap "\Delta" , which is my interpretation of for example the sentences in lines 184-185 after Thm 2. So, the rate of the convergence under your analysis of in-expectation convergence depends on how far the method is started, and can be a slow \sqrt{T} in some cases? That is new and an unusual observation.

Clarity: The english is clear, but I have a hard time reconciling this with existing literature given the lack of prior work discussion and comparison.

Relation to Prior Work: The single largest lack of this paper is in providing a concrete context w.r.t. to the vast literature at this point on analysis of SGD convergence. The authors must provide a comparison and contrast with existing literature to tell us why we should care about their new results.

Reproducibility: Yes

Additional Feedback:


Review 3

Summary and Contributions: This paper discussed the notion of biased expectation, and it's application in analysing the convergence rate of SGD. ====================== Have read the rebuttal. Raised to 7.

Strengths: This paper uses biased expectation, a parametrized statistics as a bridge, to provide a generic framework of analyzing the convergence rate of SGD under different noise, which include heavy tailed ones.

Weaknesses: I do not see major weaknesses of the paper.

Correctness: I think it's correct and the experiments are adequate.

Clarity: Yes. This paper does a very good job in the flow of formulating the theory and inducing the examples. At the beginning the notion of biased expectation is clearly defined, and the properties of it are listed clearly with explanations and compared with well known quantities, which enables reader to gain the sense of the quantity and to refer to the results easily when they read the later sections. The example section compares usual types of noise with figures, and proposes the universality of the formal proof strategy by discussing each type of noise.

Relation to Prior Work: This paper refers to previous SGD works, but they are less related to the analysis in terms of wide ranges of noise models. But it is possible that this area is unexploited.

Reproducibility: Yes

Additional Feedback: Overall the paper looks good, but I'd like to see more motivations on this topic, say, why is it important to study different noise, which type of noise do certain real world applications belong to, etc.


Review 4

Summary and Contributions: I have read the author rebuttal. As far as I can see there has been no further justification of the switch from constant stepsizes to time-varying stepsizes, which is still making the experiments hard to interpret vis-a-vis the theory. I would strongly recommend that the authors bridge this gap. The main focus of the paper is a new quantity, called the "biased expectation" which is a slight extension of the moment generating function. After showing several nice properties (e.g., decomposition for independent variables, conditional expectations, and a concentration inequality), the author switch to analysing stochastic gradient descent under various noise settings. The authors give a meta-theorem (Thm. 1) in terms of the biased expectation, and derive various bounds on the minimum squared norm of the gradient across T steps of gradient descent. The authors provide some experiments.

Strengths: Overall, I am intrigued by the simplicity and elegance of the paper and how it seems to generalise existing results. The main result, Thm. 1, in conjunction with Prop. 2 become a powerful tool with which we can get various bounds on the minimum norm gradients of stochastic gradient descent over T iterations. In particular, it allows bounds not only in expectation, but also in terms of absolute gradient magnitudes.

Weaknesses: While the "biased expectation" appears to be a powerful tool, the overall results are restricted to the gradients of the algorithm at _some_ time t in the last T iterates. While this is a common outcome of the standard analysis of SGD, it would be nice if (with some additional assumptions on f) the results could be transposed to f(x_t) or x_t within some basin of attraction. The special case of s = 0 needs much more detailed treatment. While the authors point out in the supplement that \phi is continuous at s = 0, much of the document switches between looking at s->0 or s = 0 without explanation. Assumption 1: I see that the authors need to contol ||X_t||^2 in Thm 1. (Eq. 20 in supplement) and they do with with a "variance" sigma^2. When the mean is not zero, there seems to be some overlap between this sigma^2 and the quantities being bounded in Eq. 7, 8. Is there no better way to "orthogonalise" these assumptions? The paper is a little bit disconnected from existing extensions of SGD. In particular, I would have loved to see how the conclusions relate to variance reduction methods for SGD, e.g., SVRG, etc. The experimental results are not very useful because the connections to the theory are at best tenuous. First of all, the plots show 90th percentile results, which would skew the results and may not present typical behaviour (and certainly not "expectations" as in Thm. 2). Secondly, the algorithm here uses eta_t instead of the fixed eta choices that were discussed before, so technically none of the previous theorems apply (granted, the authors claim the theory can be extended to handle decreasing eta_t, but the entire paper focusses on the fixed eta case). Finally, the plots for bounded uncentered noise seem to raise more questions than they answer and which (as far as I can see) are not answered by the current theory. Specifically, why does biased noise sometimes speed up convergence relative to unbiased noise? Does this have something to do with where the algorithm is initialised relative to the local optimum? The paper focusses on the non-convex case, but perhaps some of those features could be more clearly illustrated by focussing the simulations also on simpler convex settings (the reasoning being that all basins of attraction look convex locally).

Correctness: As far as I checked, the results look correct.

Clarity: In large part, yes, but it is still very dense. The proofs in the supplement actually aren't that long so I would encourage the authors to bring some more of that intuition into the main paper.

Relation to Prior Work: As far as I can see this is novel, but as mentioned above, it would be good to include more context on variance stabilisation methods.

Reproducibility: Yes

Additional Feedback: See weaknesses above

[Author Response · NeurIPS 2020]

First, we would like to thank all the reviewers for their valuable feedback. We sincerely believe that the document was substantially improved by taking their comments into consideration. Due to space limitations, we now address the main remarks regarding motivation, context, theoretical results and experimental setup.

**Motivations.** In the camera-ready version, we will extend the introduction to include more information on the motivations of this work. More precisely, probabilistic bounds (i.e., on quantiles) provide three advantages over in-expectation bounds: **(1)** First, they allow to consider heavy-tailed noise distributions with infinite/undefined expectation. This setting was recently shown (Zhang et al., 2020 ; Şimşekli et al., 2020) to appear when training NLP models such as BERT over large corpora, and vision models such as AlexNet on Cifar10. **(2)** As a result, SGD may present instabilities that are often solved by running the optimization multiple times (a technique refered to as *multi-start*). Our analysis in probability explains why such a method works by showing that at least 1 out of X runs of SGD will exhibit good convergence and not be disrupted by extreme noise. **(3)** Finally, our work provides a simple and unified analysis under a large class of noise assumptions. We believe that its generality and simplicity could be useful for subsequent research.

**Context and previous work.** The additional ninth page of the camera-ready version will be used to provide a more extensive discussion of previous work and how our analysis differs from these works. More specifically, we will provide optimal convergence rates for the gradient norm $\|\nabla f(x_t)\| \leq \varepsilon$ for non-convex and smooth optimization ($O(\varepsilon^{-2})$, Carmon et al., 2019) and stochastic optimization ($\tilde{O}(\varepsilon^{-2})$ up to polylogarithmic factors, Foster et. al, 2019), as well as the standard (and suboptimal) rate for SGD ($O(\varepsilon^{-4})$, e.g. [20]) when the variance is bounded. We will also provide details on the methods to obtain fast convergence in the stochastic setting: SVRG (Reddi et al., 2016), SCSG (Lei et al., 2017), SGD4 (Allen-Zhu, 2018), NEON2 (Allen-Zhu and Li, 2018), along with their convergence rates. We will also mention the existence of algorithms with convergence guarantees to a local minimum instead of a stationary point (Allen-Zhu and Li, 2018 ; Fang et al., 2019). Our analysis allows to extend SGD convergence rates to heavy-tailed distributions, as well as quantiles instead of expectations (the motivation for both is discussed in the above paragraph). In particular, we extend the work [20] that also covers the sub-exponential and in-expectation cases, but not the heavy-tailed setting (note that Assumption A2 in [20] is equivalent to $\mu_{1/\sigma^2}(\|X_t\|^2) \leq \sigma^2$). We thank the reviewers for pointing out this weakness in our submission.

**Bound on the minimum instead of current iterate.** As pointed out by Rev. 1, obtaining tight convergence rates for the iterates $\|\nabla f(x_t)\|^2$ is hard, and most non-convex analyses focus on their minimum or average over time. However, we do agree that the iterate with minimum gradient norm can be hard to find in practice, and we have thus decided to extend all our results to the average $\frac{1}{t}\sum_{i \leq t} \|\nabla f(x_t)\|^2$. This extension is direct given our proofs and is a step towards bounds on the current iterate. We will also mention in a remark that all results imply convergence of the minimum.

**Experiments.** As pointed out by most reviewers, the experiments were not sufficiently conclusive, and potentially misleading. We have thus decided to replace them by more extensive experiments on a single ML application: ridge regression with a heavy-tail (Student's t) noise distribution of tail-index $b = 1.5$. The experiments were run 1000 times in order to better approximate expectations, quantiles, and biased expectations. Figure 1 shows several aspects of the experiments: (1) The expectation of $\frac{1}{t}\sum_{i \leq t} \|\nabla f(x_i)\|^2$ reaches extremely large values (infinite in theory) compared to quantiles. (2) The choice of $\eta_t \propto t^{-1/b}$ leads to the $t^{(b-1)/b}$ convergence rate of Theorem 5. (3) Standard step-sizes $\eta_t \propto t^{-1/2}$ and $\eta_t \propto 1$ (and independent of the desired precision $\varepsilon$) lead to suboptimal convergence rates, indicating that the choice $\eta_t \propto t^{-1/b}$ may be valuable for practitioners when the noise distribution is particularly fat-tailed. (4) Biased expectations $\mu_{-s}(\frac{1}{t}\sum_{i \leq t} \|\nabla f(x_i)\|^2)$ are well aligned with theory ($s^{(b-2)/2}$ in Eq. 13).

**Additional remarks. (1)** $s = 0$ will be carefully replaced by the notation $s \to 0$. **(2)** Far from a stationary point, SGD indeed converges in $O(1/\sqrt{t})$ for $\|\nabla f(x_t)\|$ (see for example [20, Corollary 2.2] that exhibits the exact same behavior).

(a) Mean and 50%-quantiles of $\frac{1}{t}\sum_{i \leq t} \|\nabla f(x_i)\|^2$.      (b) Biased expectation $\mu_{-s}(\frac{1}{t}\sum_{i \leq t} \|\nabla f(x_i)\|^2)$.

Figure 1: SGD on a ridge regression problem with heavy-tail (Student's t) distribution of tail-index $b = 1.5$.

[Meta-Review · NeurIPS 2020]

After significant discussions with the reviewers, the reviewers were all unanimously in appreciation of the simplicity and cleanliness of the approach presented by the paper. However the authors are strongly encouraged to improve the presentation of the paper - especially the crucial proof of Lemma 1 - multiple steps have been contracted in the presentation and clarifying them is necessary. Furthermore the case of the diminishing step-size scheme is strongly suggested to be fleshed out in theory rather than being left as straightforward extensions. Lastly, the reviewers suggested to use heavier tailed distribution like the Levy distribution to verify the theory better.